# OpenReview forum: "Censoring with Plausible Deniability: Asymmetric Local Privacy for Multi-Category CDF Estimation"
_ICML.cc/2026/Conference — ICML 2026 regular_

### Official Review · Reviewer_HuLm · 2026-02-28

**Soundness:** 3
**Presentation:** 3
**Significance:** 3
**Originality:** 3
**Overall Recommendation:** 4
**Confidence:** 4

**Summary:**

This paper studies local-private estimation of multi-category sub-CDFs for a continuous numerical variable X and a categorical attribute Y under an asymmetric privacy model. The key assumption is that larger values of X are more privacy-sensitive, while Y is not private by itself but may become identifying when combined with evidence that X is large. The proposed method uses random thresholding of X, an asymmetrically censored randomized response mechanism (ACRR) that hides Y when X appears sensitive, and a current-status / competing-risks style estimation procedure with asymptotic guarantees.

**Compliance With Llm Reviewing Policy:**

Affirmed.

**Final Justification:**

My concerns have been largely addressed, and thus I updated my overall recommendation accordingly.

**Key Questions For Authors:**

1.  Can the authors provide a more explicit adversary model to formalize the joint-leakage concern? Specifically, which types of joint leakage are formally protected under the ULDP guarantee, and which are not? This would help close the gap between the privacy narrative and the formal guarantees.

2.  What happens when the monotone sensitivity assumption fails — for instance, when risk is non-monotone in X? Does the framework degrade gracefully, or does it break down entirely? Additionally, the threshold is randomly generated from a distribution G, how should practitioners choose the threshold distribution G for a given data distribution or query class?

3.  In what concrete scenarios does avoiding a predefined sensitive region offer a material practical advantage over standard ULDP with a fixed cutoff? Can the authors provide empirical or theoretical evidence under realistic settings such as user-specific cutoffs or temporal drift?

4.  Why were stronger LDP baselines, such as piecewise mechanisms, hybrid mechanisms, or discretization-plus-frequency-oracle approaches, not included in the experiments? A direct comparison is necessary to substantiate the practical advantage of the proposed method.

**Limitations:**

The paper should briefly discuss the limitations of its strong sensitivity assumptions and clarify the gap between the privacy narrative and the formal ULDP guarantees.

**Strengths And Weaknesses:**

Strengths

The paper targets a clear and potentially useful setting where privacy risk is asymmetric in a continuous attribute and joint leakage through a categorical attribute is a concern. The overall pipeline is conceptually clean: random thresholding, conditional censoring, and shape-constrained recovery fit together in a coherent way. The paper also includes substantial theoretical analysis, including consistency and pointwise asymptotic results, and the presentation is generally careful.


Weaknesses

-The problem formulation relies on a strong modeling assumption: privacy risk is monotone and one-sided in X, while Y is assumed not to be intrinsically sensitive and only becomes risky as a quasi-identifier. This assumption is hard-coded into both the thresholding construction and the censoring mechanism, making the framework narrow and potentially of limited applicability beyond this specific asymmetric privacy setting. The paper does not adequately justify why this is the right model or discuss what happens when it fails.

-A main selling point is that the method does not require a fixed sensitive cutoff in advance. However, the paper does not clearly demonstrate when this matters in practice,  for example under user-specific cutoffs, temporal drift, or threshold misspecification. Without such evidence, it remains unclear whether standard ULDP with a chosen cutoff would achieve similar performance. A direct empirical or theoretical comparison under realistic misspecification scenarios would significantly strengthen this claim.

-The proposed ACRR mechanism is explicitly characterized as a special case of utility-optimized randomized response under ULDP,  making  the privacy mechanism feel closer to an instantiation of existing ideas than a genuinely new primitive. The estimation pipeline also closely mirrors Liu et al. (2024), so the overall contribution reads more like a problem-specific extension than a fundamentally new technical solution.

-The empirical section is not fully convincing. The experiments illustrate consistency trends but do not compare against strong LDP baselines for continuous data, such as piecewise mechanism, or the most recent optimal piecewise-based mechanism, or discretization-plus-frequency-oracle approaches paired with a strategy for handling Y. As a result, the empirical section does not make a compelling case that the proposed method offers a clear practical advantage over existing alternatives.

-There are also some minor issues. The paper should more clearly specify the adversary model and which types of joint leakage are formally covered by the ULDP guarantee, since the privacy narrative currently feels somewhat disconnected from the formal guarantee on the coarsened output space. In addition, Proposition 1 would benefit from a brief proof sketch, and the ULDP notation there is slightly imprecise: the protected output should be written as a singleton set rather than a single symbol.

---

> ### Author Rebuttal · Authors · 2026-03-31
>
> Thank you for the careful review. We address each concern below.
>
> Q1/W5. We will make the adversary model explicit. We consider a Bayesian membership-inference attacker who observes the threshold $T$ and released symbol $a\in\{e_1,\ldots,e_K,e_{K+1}\}$, knows the mechanism and parameters, and may have arbitrary side information. The protected output is the singleton $\{e_{K+1}\}$. The ULDP guarantee concerns inference of the coarsened sensitive event induced by thresholding and censoring; the exact raw value of $X$ cannot be leaked by design. Writing $\pi_t=\mathbb{P}(X>t\mid t)$, the protected output satisfies $\mathbb{P}(X>t\mid a=e_{K+1},t)=e^\epsilon\pi_t/(1-\pi_t+e^\epsilon\pi_t)$, so posterior odds inflate by exactly $e^\epsilon$. By contrast, safe outputs $e_k$ intentionally reveal $X\le T$ together with $Y=k$; that leakage is outside the protected set by design. We will add this adversary model, clarify the protected target, sketch Proposition 1, and correct the singleton notation.
>
> W1/Q2. The mechanism targets the common one-sided case. If the direction is reversed, apply it to $1-X$. If sensitivity depends on distance to the boundary, use $s(X)=2|X-1/2|$ and separately collect $B=\mathbf{1}\{X>1/2\}$; if $B$ is non-sensitive it can be reported truthfully, and if sensitive it can be privatized with an additional RR step. More generally, if domain knowledge gives a one-to-one sensitivity ordering, one can relabel deterministically and map the estimated CDF back through the inverse map. These extensions are beyond our scope, but they show the framework is not tied to one ordering.
>
> Regarding $G$, the theory only requires $g(t)>0$ on $[0,1]$ for consistency. Practically, $G$ is a design distribution rather than a fragile tuning parameter. The new $G$-ablation is below (mean (std) of $L_\infty$ error; $n=10000$, $\epsilon=2$, common $F$ across all $K=4$ groups).
> |$F\backslash G$ |$x^3$|$x^2$|$x$|$x^{1/2}$|$x^{1/3}$|
> |-|-:|-:|-:|-:|-:|
> |$x^3$|**0.122 (0.016)**|0.126(0.024)|0.186(0.023) |0.357(0.035)|0.477(0.037)|
> |$x^2$|0.141(0.028)|**0.126(0.019)**|0.129(0.026)|0.251(0.031)|0.364(0.034)|
> |$x$|0.172(0.038)|0.157 (0.032)|**0.113 (0.015)**|0.127(0.025)|0.184(0.027)|
> |$x^{1/2}$|0.209(0.046)|0.186(0.038)|0.150(0.030)|0.132(0.026)|**0.117 (0.021)**|
> |$x^{1/3}$| 0.245 (0.060)|0.207(0.044)|0.174 (0.043)|0.143(0.029) |**0.126 (0.022)**|
>
> The best row-wise choice is on or near the diagonal, so a closer $G$ helps when the shape of $F$ is roughly known. Uniform $G=x$ is a robust default: it is best when $F=x$ and nearly tied when $F=x^2$ (0.129 vs. 0.126).
>
> W2/Q3. Avoiding a predefined sensitive region matters when no single cutoff is credible—for example, when users have different private cutoffs, the boundary drifts over time, or only the direction of sensitivity is known. Equivalently, ACRR averages fixed-cutoff ULDP mechanisms over random cutoffs $T\sim G$: a fixed-cutoff method commits to one analyst-chosen threshold and yields a step-function protection profile, whereas ACRR induces $\mathbb{P}(a=e_{K+1}\mid X=x)=e^{-\epsilon}+(1-e^{-\epsilon})G(x)$, a smooth monotone profile in $x$. This better matches heterogeneous cutoffs, analyst uncertainty, temporal drift, and transformed scores such as $1-X$ or $2|X-1/2|$. Our claim is not that fixed-cutoff ULDP is invalid, but that ACRR is better aligned with settings where the direction is known but the exact boundary is not.
>
> W3. We are not claiming a new privacy primitive. The novelty lies in the setting and the mechanism-estimation pair: estimating continuous $X$ together with categorical $Y$ under asymmetric privacy when only the direction of sensitivity is known and no fixed sensitive region is specified. The key design is to censor $Y$ when $X$ appears sensitive, which changes both utility and estimation. This leads to a transformed-data view, joint recovery of multiple sub-CDFs under a competing-risks structure, and a tailored asymptotic analysis. Our contribution is therefore the new problem formulation, the censoring-based mechanism class, and the accompanying theory.
>
> W4/Q4. These baselines were not included initially because none directly matches our target. Piecewise and hybrid methods privatize $X$ itself, while discretization-plus-frequency-oracle estimates a discretized joint domain. Both also require an extra choice for handling $Y$, which changes either the privacy target or the budget allocation. Our experiments therefore focused on validating the proposed estimator, while Appendix F compared analytically matched symmetric-LDP variants that isolate the effect of censoring. The point is not that those baselines are invalid, but that they solve a different privacy-estimation problem. In particular, uncensored methods must inject more noise when protecting finer distinctions inside the sensitive region. Due to space limitations, kindly refer to Reviewer vVpy, W2 for additional real data analysis.

---

> > ### Author Rebuttal · Reviewer_HuLm · 2026-04-04
> >
> > My concerns have been largely addressed, and I have updated my overall recommendation accordingly.

---

> > > ### Author Response · Authors · 2026-04-06
> > >
> > > Thank you for the careful review and follow-up. We appreciate your constructive feedback and are glad the rebuttal addressed your main concerns. We will revise the paper accordingly in the final version.

---

### Official Review · Reviewer_anFi · 2026-03-09

**Soundness:** 3
**Presentation:** 3
**Significance:** 3
**Originality:** 3
**Overall Recommendation:** 5
**Confidence:** 4

**Summary:**

This paper studies private estimation of multi-category sub-CDFs $F_{0k}(t)=\Pr(X\le t,\; Y=k), k=1,\dots,K,$
for data pairs $(X,Y)$ where $X\in[0,1]$ is continuous and larger values are considered more sensitive, while $Y$ is a categorical attribute that may become identifying when combined with sensitive $X$. The paper proposes an asymmetric local-privacy mechanism, Asymmetrically Censored Randomized Response, which samples a random threshold $T$, converts the numeric response into a binary sensitive-status indicator, and censors the category label whenever the response is potentially sensitive. This provides plausible deniability without requiring the analyst to pre-specify a fixed sensitive region on the original domain.

On the estimation side, the paper gives a clean statistical interpretation of the privatized reports as a current-status competing-risks problem, fits the distorted sub-CDFs via a nonparametric MLE/ICM procedure, and then inverts the linear distortion induced by the privacy mechanism. The paper establishes asymptotic guarantees, including consistency, cube-root pointwise weak convergence of Chernoff type, and extensions to interval-conditional predictive probabilities. It also provides finite-sample experiments, runtime studies, a divide-and-conquer implementation variant, comparisons to symmetric LDP alternatives, and a real-data demonstration.

**Compliance With Llm Reviewing Policy:**

Affirmed.

**Final Justification:**

The rebuttal adequately addressed my main concerns. In particular, the authors clarified the source of novelty—namely the ULDP mechanism and the induced nonstandard statistical structure—and explained why existing analytical tools are not directly applicable in this setting. The responses to my questions on refusal behavior, the capping step, and the divide-and-conquer procedure were clear and technically reasonable.

Overall, the rebuttal reinforces my view that the paper is technically sound and provides a coherent and meaningful contribution, even if some components build on existing tools. I therefore maintain my positive recommendation for acceptance.

**Key Questions For Authors:**

1. The definition itself is quite interesting. Taking the survey protocol in Appendix A as an example, the plausible deniability comes from some low-debt participants occasionally reporting “Yes,” thereby concealing the true high-debt participants. However, from the perspective of high-debt individuals, the only valid output is “Yes.” This asymmetry could lead to survey refusal or misreporting if respondents are not well informed about the mechanism. My question is: if such refusal occurs, can the current framework naturally accommodate it?

2. Can you say more about the capping step and the divide-and-conquer procedure in Appendix E, for example in terms of consistency, bias, or practical guidance for choosing the number of splits $M$?

3. From a nonparametric/statistical perspective, do you believe the observed $n^{-1/3}$ behavior is intrinsic to this private current-status formulation, or is there a plausible path to sharper rates (e.g., via smoothing or federated aggregation) while preserving the same privacy guarantees?

**Limitations:**

Yes

**Strengths And Weaknesses:**

- Strengths:
  - The paper is generally well structured. The motivation is easy to understand, and the progression from mechanism design to statistical reconstruction to asymptotic theory is coherent.
  - The proposed mechanism is simple, interpretable, and technically well aligned with the target estimation problem. In particular, the conditional censoring idea in Def. 4 / Fig. 1 is a natural way to protect category information only when the numeric response may be sensitive.
  - From a statistical perspective, the reduction in Sec. 3.3 is elegant: the privatized reports can be viewed through a current-status competing-risks lens, which leads to a principled estimator rather than an ad hoc reconstruction rule. Algorithm 1 makes the recovery pipeline concrete.
  - The technical development in Sec. 4 is substantial. The paper proves consistency and pointwise weak convergence for the private estimator, and extends the theory to interval-conditional category prediction. This is a nontrivial and well-motivated use of shape-constrained asymptotics under local privacy.
- Weaknesses:
  - The asymptotic toolkit naturally builds on established current-status / competing-risks theory, so the novelty lies more in the mechanism, modeling reduction, and integration than in inventing entirely new proof machinery.
- Some minor points:
  - Some places appears *Chapter xx* (e.g., line 1117), maybe it's a wrong cleveref command.

---

> ### Author Rebuttal · Authors · 2026-03-31
>
> Thank you for the thoughtful review and for recognizing the coherence of the mechanism, the current-status reduction, and the theoretical development. We have addressed each of your concerns below. Should you have any further questions, please do not hesitate to ask.
>
> W1.
> We agree that our analysis draws on existing theories. However, our novelty lies in the ULDP mechanism and the observation that sensitive outputs can be merged. In this setting, existing tools are not directly applicable because the ULDP mechanism creates privacy-induced nondifferentiable points that violate the smoothness assumptions underlying classical analyses (page 6, lines 315–324). Our contribution therefore lies in the new problem formulation, the censoring-based design, the transformed-data recovery view, and the accompanying asymptotic analysis required for this setting.
>
> W2 (presentation). Thank you for your corrections. We have carefully double-checked the manuscript and have tried to eliminate all typos and errors.
>
> Q1 (refusal or misreporting). This is a great question and gets right to the motivation for our ULDP design. The key protection comes from some non-sensitive users also producing the protected response, not from sensitive users producing a safe response. Writing $S_t=\{X>t\}$ and $\pi_t=\mathbb{P}(S_t\mid t)$, ACRR gives $\mathbb{P}(S_t\mid a=e_{K+1},t)=e^\epsilon\pi_t/(1-\pi_t+e^\epsilon\pi_t)$, so observing the protected symbol does not identify a respondent as sensitive; it only changes the prior odds by the standard $e^\epsilon$ factor. This is exactly the plausible-deniability mechanism we use to reduce refusal incentives, and we will add this Bayes argument to the manuscript. The current theory still assumes respondents follow the prescribed randomized protocol; if one wants to model systematic refusal / nonresponse explicitly, that would require an additional refusal symbol or missingness model, which we can note as a limitation.
>
> Q2 (capping). On the capping step: this is a server-side post-processing device introduced for interpretability and numerical stability, not an additional privacy mechanism, so it does not consume extra privacy budget. Appendix E.2 already notes that occasional overshoot of the recovered total CDF above 1 does not invalidate the main theoretical error bounds; the motivation for capping is instead practical. The correction only activates after the first crossing time and then freezes all sub-CDFs, so its effect is concentrated near the upper boundary. We therefore view it as a conservative boundary stabilization step, but empirically it behaves almost identical to an oracle that knows the true marginal proportions. We will revise the paper to make this distinction more explicit.
>
> On the divide-and-conquer procedure: our intent here is primarily computational. For a fixed number of splits $M$, consistency should be preserved, since each block estimator is computed on $n/M$ samples and the block size still diverges, while averaging finitely many consistent estimators remains consistent. The more delicate regime is when $M$ grows with $n$: then one trades lower runtime and possible variance reduction against increased blockwise bias. This is exactly why Appendix E.1 presents the rate discussion only heuristically and explicitly notes that the bias is difficult to characterize, while the extreme case $M=n$ is inconsistent. In practice, our recommendation is therefore to use a small fixed $M$ so that each split remains sufficiently large. In our experiments, $M\in\{2,4,8\}$ all improved over the no-splitting baseline, and $M=4$ gave a good runtime/accuracy tradeoff.
>
> The new $K$-scaling results at $n=10000$ are below (median runtime over 100 runs).
>
> | $K$ | 2 | 4 | 8 | 16 | 32 |
> |---|---:|---:|---:|---:|---:|
> | Runtime (seconds) | 0.03052 | 0.07561 | 0.12020 | 0.21539 | 0.51200 |
>
> This supports the view that even without splitting, the implementation scales mildly in $K$ at the problem sizes we study. We will add this practical guidance in the revision.
>
> Q3 (is the $n^{-1/3}$ rate intrinsic?). To the best of our knowledge, the $n^{-1/3}$ rate is currently the best known convergence rate for LDP/ULDP CDF estimation, although it is slower than the non-private rate of $n^{-1/2}$. As you correctly pointed out, sharper rates may be attainable through approaches such as smoothing techniques or federated estimators, as discussed on page 8, lines 428--438.
>
> Groeneboom, P., Jongbloed, G., and Witte, B. Maximum smoothed likelihood estimation and smoothed maximum likelihood estimation in the current status model. Annals of Statistics, 38(1):352–387, 2010.
>
> Shi, C., Lu, W., and Song, R. A massive data framework for m-estimators with cubic-rate. Journal of the American Statistical Association, 113(524):1698–1709, 2018.

---

> > ### Author Rebuttal · Reviewer_anFi · 2026-04-01
> >
> > The authors have well addressed my questions, and I therefore maintain my positive recommendation for acceptance.

---

> > > ### Author Response · Authors · 2026-04-06
> > >
> > > Thank you for the careful review and positive assessment. We are glad the rebuttal resolved your questions. We will reflect the discussed clarifications in the final version.

---

### Official Review · Reviewer_xnVZ · 2026-03-09

**Soundness:** 4
**Presentation:** 3
**Significance:** 3
**Originality:** 3
**Overall Recommendation:** 4
**Confidence:** 3

**Summary:**

This paper introduces the "Asymmetrically Censored Randomized Response" (ACRR) mechanism, a new method designed for Utility-Optimized Local Differential Privacy (ULDP) frameworks. The authors address the challenge of collecting and analyzing continuous sensitive data paired with categorical demographic information. By utilizing a binary encoding, the mechanism introduces an inherent asymmetry in sensitivity.  The paper provides theoretical guarantees for uniform consistency and pointwise weak convergence, and validates these through numerical experiments.

**Compliance With Llm Reviewing Policy:**

Affirmed.

**Key Questions For Authors:**

1.	The binary encoding relies on a threshold distribution $G$ (e.g., uniform). If the true distribution of the sensitive attribute $X$ is highly skewed or concentrated in a narrow range, how does the choice of $G$ affect the estimation error? Is there a theoretical "optimal" distribution $G$ for a given (unknown) distribution of $X$?
2.	Could the authors provide a more detailed discussion on the computational complexity of the MLE approach via the ICM algorithm, especially as the number of categories $K$ grows large? Does the memory usage or runtime become a bottleneck in large-scale deployments?
3.	While censoring categorical information for sensitive $X$ improves privacy, does this significantly impact downstream stratified inference tasks (e.g., if one needs to perform subgroup analysis precisely for those with sensitive $X$)?

**Limitations:**

Yes.

**Strengths And Weaknesses:**

•	Soundness:  The approach is theoretically grounded in the ULDP framework, and the authors derive asymptotic properties ($L_2$ and $L_\infty$ consistency) for their CDF estimator, which aligns with existing literature on LDP mechanisms .
•	Presentation: The paper is clearly written and well structured.
•	Significance:  The contribution is practical for survey data collection where users may be willing to disclose certain non-sensitive attributes but are reluctant to provide full, identifiable profiles . While the authors note that it can be applied to symmetric LDP settings, the primary gains appear to be tied to the specific constraints and definitions of ULDP.
•	Originality: The idea of using deterministic binary encoding for threshold comparison, combined with asymmetric censoring, is an elegant and novel way to bypass the rigidity of defining sensitive regions in numerical attributes.

---

> ### Author Rebuttal · Authors · 2026-03-30
>
> Thank you for the thoughtful review and for recognizing the novelty of combining threshold encoding with asymmetric censoring. We have addressed each of your concerns below. Should you have any further questions, please do not hesitate to ask.
>
>
> Q1. (Choice of $G$). According to our theory, the minimum variance is achieved when $F = G$, as shown in Theorem 3. However, since the distribution $F$ is unknown in practice, it is impossible to set $F = G$ exactly. A naive alternative is to use $\hat{F}$ as the sampling distribution to construct a refined estimator. Nevertheless, such an approach requires splitting either the sample size or the privacy budget, and therefore we do not recommend it. Regarding your concern that the distribution of $X$ may be highly skewed or concentrated on a narrow range, Theorem 2 shows that uniform consistency still holds as long as the density of $G$ is strictly positive on $[0,1]$.
>
> The new $G$-ablation is below (mean (std) of $L_\infty$ error; $n=10000$, $\epsilon=2$, common $F$ across all $K=4$ groups).
>
> | $F \backslash G$ | $x^3$ | $x^2$ | $x$ | $x^{1/2}$ | $x^{1/3}$ |
> |---|---:|---:|---:|---:|---:|
> | $x^3$     | **0.122 (0.016)** | 0.126 (0.024) | 0.186 (0.023) | 0.357 (0.035) | 0.477 (0.037) |
> | $x^2$     | 0.141 (0.028) | **0.126 (0.019)** | 0.129 (0.026) | 0.251 (0.031) | 0.364 (0.034) |
> | $x$       | 0.172 (0.038) | 0.157 (0.032) | **0.113 (0.015)** | 0.127 (0.025) | 0.184 (0.027) |
> | $x^{1/2}$ | 0.209 (0.046) | 0.186 (0.038) | 0.150 (0.030) | 0.132 (0.026) | **0.117 (0.021)** |
> | $x^{1/3}$ | 0.245 (0.060) | 0.207 (0.044) | 0.174 (0.043) | 0.143 (0.029) | **0.126 (0.022)** |
>
> The best row-wise choice is on or near the diagonal, so a closer $G$ helps when the shape of $F$ is roughly known. At the same time, uniform $G(x)=x$ is a robust default: it is best when $F=x$, nearly tied when $F=x^2$, and avoids severe misspecification. We therefore view $G$ as a design distribution rather than a fragile tuning parameter: use a closer shape if prior knowledge is available, and otherwise use uniform.
>
>
> Q2. (computational complexity as $K$ grows). The ICM step operates on the ordered thresholds and jointly updates $K$ sub-CDFs under the shared constraint through $F_+(t)$. With $m$ unique thresholds ($m\le n$), a straightforward implementation stores $\mathcal{O}(Km)$ function values, while the raw one-hot data remain $\mathcal{O}(n)$. The new $K$-scaling results at $n=10000$ are below (median runtime over 100 runs).
>
> | $K$ | 2 | 4 | 8 | 16 | 32 |
> |---|---:|---:|---:|---:|---:|
> | Runtime (seconds) | 0.03052 | 0.07561 | 0.12020 | 0.21539 | 0.51200 |
>
> The runtime is close to linear in $K$. Memory is also linear in both $K$ and sample size $N$; for the main matrices it is about $((72K+160)N)$ bytes. All experiments fit comfortably within 1GB of RAM, so memory is not a practical bottleneck at the scales we study. We will add these numbers and clarify that the current implementation is intended for small-to-moderate $K$.
>
>
> Q3. (downstream stratified inference in sensitive $X$). If we understand your question correctly, we believe that once users obtain an LDP estimate of $X$, subsequent inference for $X$ can be carried out directly while preserving privacy protection. However, unlike in the non-private setting, the convergence rate is typically slower than $n^{-1/2}$, which may affect the performance of the final inferential procedures.
>
> We add that, in our setting, the recovered sub-CDFs allow us to estimate $\mathbb{P}(Y=k \mid X\in I)$ for any interval $I$, including sensitive regions such as $X>u$. A representative sensitive-region result is below for the target $\mathbb{P}(Y=k, X>1/2)$ at $\epsilon=1$.
>
> | $n$ | $10^3$ | $10^5$ | $10^6$ |
> |---|---:|---:|---:|
> | Prediction error | 0.138 | 0.031 | 0.013 |
>
> So subgroup analysis in the sensitive region is harder, but it is not ruled out; censoring trades per-user label disclosure for consistent population-level subgroup inference.

---

> > ### Author Rebuttal · Reviewer_xnVZ · 2026-04-03
> >
> > My concerns have been mainly addressed.

---

> > > ### Author Response · Authors · 2026-04-06
> > >
> > > Thank you for the thoughtful review and follow-up. We are pleased that our responses addressed your concerns. We will incorporate these clarifications in the final version.

---

### Official Review · Reviewer_vVpy · 2026-03-14

**Soundness:** 4
**Presentation:** 4
**Significance:** 3
**Originality:** 3
**Overall Recommendation:** 4
**Confidence:** 3

**Summary:**

This paper studies distribution estimation under ULDP for data with one continuous X and one categorical Y. The objective is to estimate the category-wise joint CDFs $F_{0k}(t)$ under an asymmetric privacy setting.
A key choice is that the method only assumes a sensitive direction, rather than fixing a sensitive region in advance. Based on this, the authors binarize X using a random threshold and design an asymmetrically censored randomized response mechanism.
On the estimation side, they connect the privatized observations to a current-status competing risks problem and then invert the privacy map to recover the target CDFs. The paper also develops a fairly complete theoretical analysis, including consistency and pointwise asymptotic guarantees, and supports the method with synthetic data and a small real-data study.

**Compliance With Llm Reviewing Policy:**

Affirmed.

**Final Justification:**

I appreciate the authors’ detailed response. Most of my concerns have been fully addressed, so I will maintain my positive assessment.

**Key Questions For Authors:**

1. Can the authors provide stronger empirical comparisons against the closest continuous-data LDP/ULDP baselines under matched privacy budgets?
2. Add the sensitive ablation on the threshold distribution G.
3. The current framework assumes the sensitive direction is known a priori. Can the authors discuss whether the approach could be extended to settings where sensitivity is non-monotone in X, or where the sensitive direction is unknown?
4. Since label information is censored exactly in the sensitive region, can the authors clarify under what sample size or privacy regime the estimation in that region is still reliable in practice?

**Limitations:**

yes

**Strengths And Weaknesses:**

**Strengths**

- The paper addresses a realistic gap between full LDP and application needs: in many cases, privacy sensitivity is directional rather than defined by a fixed interval. I think this motivation is reasonable.
- The proposed ACRR mechanism is easy to understand. The idea of conditionally censoring the category label when the continuous value looks sensitive is intuitive. And the connection to competing-risks current-status estimation is nice, which brings strong technical insights.
- The theory part is solid. The paper provides more than a privacy definition and consistency; there are also asymptotic results.

**Weakness**

- The experiments mostly show that the proposed method works as expected, but they do not fully convince me that this method is clearly better than the closest existing ULDP alternatives. In particular, strong head-to-head baseline comparison is still limited.
- The real-data validation is weak. It is acceptable as a sanity check, but not strong enough to demonstrate practical impact in a realistic privacy-sensitive setting. Adding more real-data experiments would enhance the draft quality.
- The paper does not fully explore how sensitive performance is to the choice of threshold distribution G.

---

> ### Author Rebuttal · Authors · 2026-03-31
>
> Thank you for the thoughtful review. We address each concern below.
>
> Q1 / W1. Direct head-to-head comparisons would be useful, but the nearest off-the-shelf baselines are not exact matches because the privacy target differs: our mechanism protects sensitive-group membership while censoring $Y$, whereas the closest continuous-data LDP / ULDP baselines estimate $X$ itself or the full uncensored state $(\mathbf{1}_{X>T},Y)$. As noted in Appendix F, these uncensored baselines therefore perturb more strongly and incur the $K$-dependent inflation that censoring avoids.
>
> For membership inference, however, the protection is on the same Bayes-factor scale. Writing $\pi_t=\mathbb{P}(X>t\mid t)$, ACRR gives $\mathbb{P}(X>t\mid a=e_{K+1},t)=e^\epsilon\pi_t/(1-\pi_t+e^\epsilon\pi_t)$, so the posterior odds are exactly $e^\epsilon$ times the prior odds. Thus censored ULDP offers comparable protection for inferring sensitive-group membership while avoiding the extra perturbation needed to protect finer distinctions in the sensitive region. Appendix F is therefore the most apples-to-apples comparison in the current paper, and we will revise the wording to avoid any claim of universal empirical dominance over methods designed for a different privacy target.
>
> W2. We agree that the current fairadapt study is best viewed as a sanity check, not the main practical evidence, and we will revise the wording accordingly. We also add a new analysis on an ACS PUMS–derived income–region dataset built from public U.S. Census microdata, with up to 750,017 observations in our four-state extract. Due to space limitations, we present only findings on the capping mechanism here; see [supplementary material](https://anonymous.4open.science/r/anonymous_rebuttal_figures-28AC/supplementary_figures.pdf). The full analysis is included in the revised manuscript. The figures show that the estimated errors of “stop at one” and “oracle” are close and their distributions are nearly identical, while “uncorrected” shows substantial bias, indicating that capping effectively reduces error.
>
> Q2 / W3. Our theory guarantees consistency as long as $G$ has positive density on $[0,1]$. In practice, $G$ is a design choice: if one roughly knows the shape of $F$, a closer $G$ usually helps; otherwise uniform is a strong default. The new $G$-ablation is below (mean (std) of $L_\infty$ error; $n=10000$, $\epsilon=2$, common $F$ across all $K=4$ groups).
>
> |$F \backslash G$|$x^3$|$x^2$| $x$|$x^{1/2}$|$x^{1/3}$|
> |-|-:|-:|-:|-:|-:|
> | $x^3$     | **0.122 (0.016)** | 0.126 (0.024) | 0.186 (0.023) | 0.357 (0.035) | 0.477 (0.037) |
> | $x^2$     | 0.141 (0.028) | **0.126 (0.019)** | 0.129 (0.026) | 0.251 (0.031) | 0.364 (0.034) |
> | $x$       | 0.172 (0.038) | 0.157 (0.032) | **0.113 (0.015)** | 0.127 (0.025) | 0.184 (0.027) |
> | $x^{1/2}$ | 0.209 (0.046) | 0.186 (0.038) | 0.150 (0.030) | 0.132 (0.026) | **0.117 (0.021)** |
> | $x^{1/3}$ | 0.245 (0.060) | 0.207 (0.044) | 0.174 (0.043) | 0.143 (0.029) | **0.126 (0.022)** |
>
> The best entry in each row is on or near the diagonal. Thus matching $G$ to the rough shape of $F$ helps; otherwise uniform is a strong default.
>
> Q3. The current mechanism is designed for the common case where the sensitive direction is known and one-sided. If the direction is reversed, the same construction can be applied to $1-X$. If sensitivity depends only on distance to the boundary, one may use $s(X)=2|X-1/2|$ as the new sensitivity variable and separately collect the side bit $B=\mathbf{1}\{X>1/2\}$; if $B$ is non-sensitive it can be reported truthfully, and if $B$ is sensitive it can be privatized with an additional RR step. More generally, if domain knowledge gives a one-to-one ordering by sensitivity, one can relabel deterministically and then map the estimated CDF back through the inverse map. For example, after partitioning $[0,1]$ into ten bins, one could reorder them according to a known sensitivity ranking such as $5,6,4,7,3,8,2,9,1,10$ before thresholding. These extensions are beyond the main scope, but they show that the framework is not tied to one literal ordering.
>
> Q4. There is no universal sample-size cutoff, because reliability depends on $n$, the privacy level $\epsilon$ through the factor $1/(1-e^{-\epsilon})$, and the rarity of the category or tail region of interest. Empirically, inference in the censored region is harder but still improves steadily with $n$ and $\epsilon$. A representative sensitive-region result is below for the target $\mathbb{P}(Y=k, X>1/2)$ at $\epsilon=1$.
>
> |$n$|$10^3$|$10^5$| $10^6$ |
> |-|-:|-:|-:|
> | Prediction error | 0.138 | 0.031 | 0.013 |
>
> The corresponding errors are smaller for $\epsilon=2,3$. We will add this discussion to clarify that sensitive-region inference is feasible, but it requires larger effective sample sizes than inference on $X\le u$.

---

> > ### Author Rebuttal · Reviewer_vVpy · 2026-04-03
> >
> > I appreciate the authors’ detailed response. Most of my concerns have been fully addressed, so I will maintain my positive assessment.

---

> > > ### Author Response · Authors · 2026-04-06
> > >
> > > Thank you for the thoughtful review and follow-up. We are glad the rebuttal addressed your concerns. We will incorporate the requested clarifications and revisions in the final version.

---

### Decision · Program_Chairs · 2026-04-30

**Decision:**

Accept (regular)

**Comment:**

This paper introduces a mechanism for releasing data under the utility-optimized differential privacy (ULDP) framework. The core idea is to first map a sensitive numeric variable to a binary bit, based on whether the variable is above or below a user-specific threshold, which is drawn from some distribution G. Then, the paper applies a ULDP mechanism to the binarized sensitive value, as well as a corresponding categorical variable.

The reviewers appreciated many aspects of this work, including the motivation for the work, the theoretical analysis, and the presentation. The reviewers did raise a few questions regarding the generality of the framework and the empirical evaluation; these issues were largely settled during the author-reviewer discussion, and the reviewers overall found the contribution to be novel and theoretically interesting.

One question from my own understanding of the work is regarding the privacy guarantees. The authors state that "it is important to note that the ULDP guarantee applies to the space of E rather than the original continuous domain". It is not entirely clear what this means. For example, since G can be any distribution, if G were a delta function such that every client's threshold $\tau_i=0.999$, then the 1-bits for thresholded-X seem to leak more about the initial continuous values than if the thresholds $\tau_i =0$. However, from the standpoint of this paper, there is no difference between the privacy guarantees of these two settings, as long as $\epsilon$ is the same. It would be helpful to clarify in more detail the nuances (and possible limitations) of this privacy analysis.